# Oligodendrocyte Dysfunction in Tauopathy: A Less Explored Area in Tau-Mediated Neurodegeneration

**DOI:** 10.3390/cells13131112

**Published:** 2024-06-27

**Authors:** Moumita Majumder, Debashis Dutta

**Affiliations:** 1Department of Microbiology and Immunology, Medical University of South Carolina, Charleston, SC 29425, USA; majumdem@musc.edu; 2Department of Pediatrics, Darby’s Children Research Institute, Medical University of South Carolina, Charleston, SC 29425, USA

**Keywords:** MAPT, oligodendrocytes, myelin, tauopathy, neurodegeneration

## Abstract

Aggregation of the microtubule-associated protein tau (MAPT) is the hallmark pathology in a spectrum of neurodegenerative disorders collectively called tauopathies. Physiologically, tau is an inherent neuronal protein that plays an important role in the assembly of microtubules and axonal transport. However, disease-associated mutations of this protein reduce its binding to the microtubule components and promote self-aggregation, leading to formation of tangles in neurons. Tau is also expressed in oligodendrocytes, where it has significant developmental roles in oligodendrocyte maturation and myelin synthesis. Oligodendrocyte-specific tau pathology, in the form of fibrils and coiled coils, is evident in major tauopathies including progressive supranuclear palsy (PSP), corticobasal degeneration (CBD), and Pick’s disease (PiD). Multiple animal models of tauopathy expressing mutant forms of MAPT recapitulate oligodendroglial tau inclusions with potential to cause degeneration/malfunction of oligodendrocytes and affecting the neuronal myelin sheath. Till now, mechanistic studies heavily concentrated on elucidating neuronal tau pathology. Therefore, more investigations are warranted to comprehensively address tau-induced pathologies in oligodendrocytes. The present review provides the current knowledge available in the literature about the intricate relations between tau and oligodendrocytes in health and diseases.

## 1. Introduction

The microtubule-associated protein tau (MAPT) is a microtubule-binding protein that plays an instrumental role in microtubule assembly and functioning and is mostly expressed in the central and peripheral nervous system. The protein was first identified in 1975 by Weingarten and co-workers [1], although the disease relevance of tau was reported almost a decade after its discovery [2]. Extensive research on tau later confirmed it to be the main component of the neurofibrillary tangles (NFTs) found in the brains of Alzheimer’s disease (AD) patients [3,4,5,6,7]. At present, there are more than 20 distinct and overlapping neurodegenerative diseases where tau inclusions are observed in different brain areas, making this protein a common pathological hallmark in these diseases, collectively called tauopathies. The most prevalent tauopathy is AD, whereas other tauopathies include frontotemporal dementia (FTD), chronic traumatic encephalopathy (CTE), progressive supranuclear palsy (PSP), corticobasal degeneration (CBD), Pick’s disease (PiD), globular glial tauopathy (GGT), argyrophilic grain disease (AGD), etc. Interestingly, in the majority of these diseases tau aggregation and associated neurodegeneration happen sporadically and progressively with aging. In contrast, there are pathological tau mutations that have been shown to be a contributory cause behind tau aggregation in the brain. The first reported mutation was a dominantly inherited form located in the MAPT gene of chromosome 17 found in frontotemporal dementia associated with parkinsonism (FTDP17) [8,9,10,11]. Till now, 65 tau mutations have been identified in different tauopathies cumulatively. The importance of tau mutations has been established by different animal studies, where presence of different tau mutations has shown age-dependent progressive tau spreading, neuronal loss, and significant cognitive decline in rodents [12,13,14]. Moreover, these experimental models also recapitulate pathological tau spreading from one brain region to another through the physiologically connected neuronal network in the brain, as speculated in human patients at different stages of clinical progression based on neuropathological analyses [15,16]. In the case of AD, tau deposition first appears in the layer II and IV of the entorhinal cortex, and spreads progressively towards the outer molecular layer of the dentate gyrus (DG) through the Perforant pathway [17,18,19,20]. Progressively, tau species spread to the CA3 region, and then, through the Schaffer-collateral pathway extend to the hippocampal pyramidal neurons of CA1 followed by prominent aggregation in the neocortex, and at that time point, patients present significant memory/cognitive deficits, marked as the Braak stage V-VI of AD disease progression [17,18,21]. It is noteworthy that tau pathology correlates well with clinical manifestations in AD patients, and therefore, targeting tau by therapeutic means is believed to be a rational approach to prevent neurodegeneration and cognitive impairments in AD patients [22,23,24]. However, as tau is present in multiple isoforms in the brain and affects different cell types depending on its structure and varied aggregated forms, the extraneuronal effect of tau aggregation should also be seriously considered for understanding the diverse tau pathology in the brain. In that respect, astrocytes and oligodendrocytes are the two other brain cells that are known to accumulate the strain-specific form of tau aggregates in the diseased brains of multiple neurodegenerative diseases. This review particularly focuses on the effect of tau on oligodendrocyte function in normal healthy and diseased brains and discusses the importance of tau–oligodendrocyte interaction, pivotal in brain function.

## 2. Structural Variants of Tau

Although there is only one MAPT gene present in chromosome 17 of the human genome (17q21.31), the pre-mRNA of tau undergoes alternative splicing to generate a total of six isoforms of the tau protein in the human brain. At the gene level, there are 16 exons flanked by intronic sequences in the MAPT gene [25,26,27]. However, the differences in the total length of the tau protein are caused by inclusion or exclusion of exon 2, exon 3, and exon 10, resulting in formation of six isoforms of differing amino acid residues, as shown diagrammatically in Figure 1. A full-length tau protein is composed of 441 amino acid residues and has a proline-rich domain towards the N-terminal, and repeat domains enriched with hydrophobic amino acids towards the C-terminal [28]. The inclusion of both exon 2 and exon 3 leads to formation of 2N tau, whereas exclusion of exon 2 or 3 or exclusion of both these exons results in formation of 1N and 0N tau isoforms (Figure 1). On the other hand, at the C-terminal region a total of four repeat domains are found in the full-length tau, but exclusion of exon 10 generates the 3R form of tau (Figure 1). Therefore, a total of six structural variants of monomeric tau, spanning from 352 to 441 amino acids, are found in the adult human brain, including 0N4R, 1N4R, 2N4R and 0N3R, 1N3R, and 2N3R (Figure 1) [29,30]. Certainly, the N-terminal and C-terminal regions of tau have distinct functions in the cellular environment. The N-terminal region of tau is thought to interact with the cell membrane and its components [31]. The PXXP motif present in the N-terminal proline-rich domain is the recognition site for SH3-domain-containing tyrosine kinases such as Fyn kinase [32]. On the other hand, the repeat domains and the surrounding flanking regions mediate its interaction with the microtubule components [33,34]. Interestingly, individual repeat domains can also bind the single microtubule protofilaments strongly, but in the absence of flanking regions these domains fail to mediate polymerization or cross-linking of microtubule protofilaments to form the microtubule assembly (Mandelkow EM et al., 1996). The existence of the 3R and 4R forms of tau is evolutionary different among vertebrates. The adult human brain contains both the 3R and 4R forms of tau in a 1:1 ratio, although the N-terminally different tau isoforms (0N, 1N and 2N) are expressed at different ratios [27,35]. In contrast, the adult mouse brain expresses only the 4R form of tau throughout the CNS, and the adult chicken brain has 3R, 4R, and 5R forms of tau [36,37]. Apart from these isoforms, there is another bigger isoform of tau, called Big-tau, containing another large exon coding the N-terminal region, in the peripheral nervous system [38,39]. The physiological function of tau is also impacted significantly by several post-translational modifications such as phosphorylation, acetylation, ubiquitination, nitration, sumoylation, methylation, glycation, and truncation. Among these modifications, phosphorylation of tau has gained relatively greater importance due to its role in facilitating dissociation from the microtubule and intermolecular self-aggregation. In fact, numerous studies have shown that the phospho-tau level is drastically higher in AD-brain-derived tau than in control brains, with a value of 8 mol of phosphates per mole of protein [40]. Varied forms of post-translational modifications of tau along with the causing proteins/enzymes are described in Table 1. 

## 3. Tau Aggregation in Tauopathies

In familial tauopathies, specific mutations in the MAPT gene are known to trigger tau aggregation. Until now, nearly 60 mutations of the MAPT gene have been reported in tauopathies including FTDP-17, PSP, and CBD [81]. However, tau mutations have not been identified in AD patients, confirming that tau aggregation also happens sporadically with aging and raising the question how tauopathy occurs in the absence of mutations. In fact, aggregated tau is found in the brains of primary-age-related tauopathy (PART), where mild or no cognitive dysfunction is manifested [82,83]. Most of these mutations occur in the microtubule-binding repeat regions and the adjacent regions such as G272V, N279K, ΔK280, P301L, V337M, and R406W [27,84,85,86]. These single-amino acid changes in the tau sequence decrease its propensity to bind the microtubule assembly and simultaneously potentiate the self-aggregation of tau. In addition, mutations occurring in exon 10 of MAPT coding for the R2 repeat region lead to inclusion of R2 in the tau structure, resulting in generation of more 4R tau than 3R, and thereby, increasing the ratio of 4R to 3R. In contrast, mutations like ΔK280, L266V, and G272V inhibit inclusion of exon 10 in MAPT mRNA, leading to a decrease in the 4R to 3R tau ratio [27]. Tau is natively unfolded and has lesser secondary structure, but aggregated tau adopts varied forms of conformations that have been extensively identified in the recent past by the advancement of cryo-electron microscopy (cryo-EM) [87]. Initial studies identified two important hexapeptide sequences present in the R2 and R3 regions (VQIINK and VQIVYK), the interactions of which comprise the first step towards formation of the β-sheet structure of tau aggregates and forms the core of assembly [88,89]. However, disease-specific tau aggregates are distinguished by the differential mixing of the 4R and 3R isoforms of tau in the larger paired helical filament (PHF) structure, and therefore, the orientation of β-sheets and folds of repeat domains in tau have been found to differ substantially among tauopathies. For example, a cryo-EM study performed at 3.4–3.5 Å resolution has demonstrated that the filament core of tau filaments isolated from human AD brains is comprised of residues 306–378, which adopt a crosslinked β-sheet structure and form the seed for tau aggregation, whereas the PHFs and straight filaments (SFs) differ in their intermolecular protofilament packing. The filament core mainly consists of the R3 and R4 sequence of tau and a few amino acids following the R4 region [30,90]. Irrespective of the presence or absence of R2, the tau isoforms can aggregate with each other to form tau filaments consisting of both 3R and 4R tau. This common structure of tau filaments was found in tau filaments isolated from different AD cases and in the filaments isolated from multiple brain regions, suggesting a signature mode of tau filament assembly in AD. Moreover, the current studies emphasize the importance of the structurally disordered flanking regions of the repeat regions in determining different conformational polymorphism of the tau aggregates [91]. On the other hand, Falcon and colleagues performed cryo-EM based structural characterization of tau filaments isolated from the frontal cortex of a patient variant of FTD containing abundant Pick bodies composed of 3R tau filaments. They reported that Pick body tau filaments are distinguished by the presence of narrow and wide Pick filaments (NPF and WPF). These filaments differ in width, where an NPF has a width of 50–150 Å and a WPF has a width of 50–300 Å. In fact, two NPF protofilaments are joined at the distal tips to form a WPF, although both NPF and WPF contain equally distanced helical crossovers along the length of the protofilament. Structurally, the NPF of Pick body tau filaments are different from the protofilaments of PHF and SF of AD tau. The core filament is stretched from residues K254-F378 of 3R tau and composed of nine β-strands linked with each other by turns, forming four stacks. Further analysis demonstrated that the core of the Pick tau protofilament is made of the R1, R3, and R4 regions and some amino acids downstream of R4, in contrast to the AD tau filament core, which is formed by the R3 and R4 regions and 10 amino acids following the R4 region. In addition, Pick’s tau filaments were found to be non-phosphorylated at S262 and/or S356 for unknown reasons [92]. Interestingly, the typical 4R tauopathies like CBD, where tau filaments are composed of only the 4R form of tau are structurally much different from 3R-only tau filaments like Pick’s disease or 3R + 4R tauopathy like AD and CTE. The core of tau protofilaments found in the 4R tauopathy is made of residues spanning from the last amino acid of R1, the whole of R2, R3, and R4 and 12 amino acids after the R4 region (K274-E380 of full-length tau), where a total of 11 β-sheet secondary structures of the protein are stacked together in four layers and connected to each other by turns and arcs. Overall, CBD tau filaments have two structural components, called narrow and wide tau filaments, which differ in the width of the filament structure [87,93]. These are called type I and type II CBD tau filaments. It should be noted that it is only in 4R tauopathies, like in CBD tau, that the core of the protofilament has the whole R2 region taking part in forming the β-sheet structure, and that this is absent in both the 3R tauopathy (PiD) and 3R + 4R tauopathy (AD and CTE), and therefore, the CBD tau folds are the longest in length, covering 107 residues, and the aggregates are characteristically observed in both neurons and astrocytes of the frontal cortex and basal ganglia [94]. In summary, these reports on the structural elucidation of varied and overlapping forms of tauopathies suggest that the presence or absence of specific repeat domains near the C-terminal end of tau determines the overall macromolecular structure of tau filaments in the brain and structural variability of tau may decide the cell-type-specific vulnerability of brain cells in tauopathies, and this unclear territory of tau pathology requires further investigations.

## 4. Tau Expression in Oligodendrocytes

In the central nervous system (CNS), tau is majorly expressed in neurons and numerous studies to date have disclosed the function of neuronal tau in normal physiology and in neurodegenerative diseases. Interestingly, studies have examined tau expression in different glial cells, where tau expression was not found in microglia and astrocytes. However, tau is shown to be expressed in oligodendrocytes, where it plays instrumental roles in microtubule assembly and other functions. Moreover, the intracellular distribution of tau mRNA and protein in oligodendrocytes was found to be similar [95]. The expression pattern of tau isoforms differs remarkably between oligodendrocytes and neurons from the neonatal to adult phase. Research conducted in rat brains of 1-day- and 20-day-old animals demonstrated that where oligodendrocytes from 1-day-old rat brain express tau with no N-terminal insert (0N tau), at the 20-day-old stage oligodendrocytes in the brain contain equal amounts of the 0N and 1N forms of tau. In contrast, cortical neurons from the 20-day-old rat brain express comparable amounts of different amino-terminally modified tau, including 0N, 1N, and 2N. A similar trend of discrepancy was also found in the carboxy-terminal modified forms of tau between oligodendrocytes and neurons at these two developmental phases. In contrast to neurons in rat cortex, which express only the 4R form of tau, older oligodendrocytes contain both the 3R and 4R forms of tau, where the 4R form is predominant. This indicates that post-transcriptional regulation of tau is different during development between neurons and oligodendrocytes, with potential implications on the functionality of these two divergent brain cells [96].

The prime function of oligodendrocytes is to produce myelin, wrapping the neuronal axons, to facilitate the conduction of currents. In this process, synthesis of myelin by oligodendrocytes is greatly impacted by tau, as studies have shown that the +PXXP motif of tau protein present in the N-terminal proline-rich domain is vital for its binding to the Fyn kinase through its SH3 domain, and that promotes oligodendrocyte processes elongation/maturation and the formation of myelin (Figure 2). In contrast, disruption of tau binding with Fyn kinase results in shortening of oligodendrocyte processes and myelin generation (Figure 2). Sorting of myelin proteins in oligodendrocytes is dependent on raft formations in the trans-Golgi network that facilitates the secretion of myelin outside the cells [97]. However, tau deficiency compromises the raft formation and the downstream myelin protein secretion by oligodendrocytes, resulting in reduced myelin formation. In fact, transport of mRNAs of mature oligodendrocyte markers and the myelin forming protein, called myelin basic protein (MBP), is significantly dependent on microtubules. Downregulation of tau or abnormal phosphorylation was found to prevent the transport of MBP mRNAs towards the periphery, and thereby, the translation of MBP protein is disrupted, resulting in impaired contacts of the oligodendrocytes processes with the neuronal axons [98] resulting in reduced myelination of neurons (Figure 2) when oligodendrocytes were cocultured with neurons derived from dorsal root ganglions of newborn rats [98]. In cultured oligodendrocytes, tau expression was shown to be adjacent to MBP in the membrane expansions and at the end of processes. Tau protein expression follows the pattern of mRNA expression in oligodendrocytes [95]. Therefore, in the absence of tau, transformation of oligodendrocyte precursor cells (OPCs) into mature oligodendrocytes is inhibited concomitant with persistent expression of the oligodendrocyte precursor cell (OPC) marker proteoglycan NG2, and that reinstates the pivotal function of physiological tau in mature oligodendrocyte formation and functioning [99].

### 4.1. Tau and Oligodendrocytes in Diseased Brains: Perspective from Animal Studies

Multiple experimental studies have confirmed that tau plays a vital role in microtubule assembly and transport of cellular materials in oligodendrocytes. This raises the question: what happens to the endogenous tau expressed exclusively by oligodendrocytes in diseased brains. Tau inclusions in oligodendrocytes are found in the majority of tauopathies such as PSP, CBD, PiD, AGD, FTDP-17, and GGT, with the exception of AD [100,101,102], where tau aggregates are predominant in the neurons. Over the years, significant attention has been paid to neuronal mechanisms of tauopathy, whereas the role of oligodendrocytes in tauopathy-associated neurodegeneration has been less explored. As stated earlier, tau expression is indispensable for microtubule functioning in oligodendrocytes and transport of myelin constituents to cellular processes for optimum arborization and oligodendrocyte maturation [99]. In contrast, abnormal over-expression of tau is also detrimental for the viability of oligodendrocytes, as evidenced by in vitro studies showing the death of cultured oligodendrocytes isolated from transgenic mice over-expressing wild-type or P301L-mutated human tau under the CNP promoter [103]. In addition, tau is hyperphosphorylated under a disease scenario by multiple kinases such as glycogen synthase kinase-3β (GSK-3β) and MAP kinases. As phosphorylation of tau in its different domains reduces its ability to bind the microtubules, it thereby exerts toxic effects on oligodendrocytes’ function and survival. 

Studies conducted in human postmortem PSP brains also showed the presence of MAPT transcripts in the nucleus and cytoplasm of neurons and oligodendrocytes as well as astrocytes by employing RNAscope coupled with light microscopy-based experiments [104]. Moreover, the content of the MAPT transcript in both neurons and oligodendrocytes in PSP brains may differ depending on the brain region. In the case of oligodendrocyte-specific MAPT expression, the highest transcript was found in the frontal white matter and lowest in the cerebellar white matter, pointing to the region-specific vulnerability of the brain in tauopathy [104]. Oligodendrocyte-specific tau aggregation is thought to be much different from astrocytic tau inclusions, although both these cell types are known to develop distinct tau aggregates in the forms of fibrils and coiled bodies, respectively, in CBD and PSP. An interesting study conducted in primary astrocytes and oligodendrocytes isolated from rat brain showed that tau can form inclusion bodies in oligodendrocytes even in the absence of neurons, whereas astrocytes need the presence of neurons in the coculture setup to develop intracellular tau aggregates [105]. The same study explored the contribution of oligodendrocytes in tau spreading in mouse brain by using neuron-specific tau-deficient mice receiving stereotaxic injection of CBD and PSP brain-derived tau into the hippocampus. Surprisingly, the authors found that in the absence of neuronal tau, visible tau pathology develops progressively in the oligodendrocytes of the fimbria and corpus callosum, regions rich in oligodendrocytes. Moreover, the oligodendrocyte-specific tau aggregation spread to the contralateral non-injected site of the brain progressively. However, this form of tau transmission in the astrocytes of the contralateral side was not observed in the neuronal tau-deficient mice [105]. This indicates that oligodendrocytes employ distinct mechanisms to transfer tau seeds through interconnected regions of the brain. It is noteworthy that wild-type mice containing endogenous tau did not develop any glial tau aggregates when injected with tau derived from AD human brain (AD-tau) but developed prominent astrocytic and oligodendroglial tau inclusions with specific structures after injection with CBD- and PSP-tau [106]. This finding suggests that the isoform-dependent formation of tau aggregates might determine the vulnerability of different brain cells in varied forms of tauopathy, and accordingly, these aggregates are transmitted to the interconnected regions, affecting the signature cell types [107], where AD-tau intracerebral injection leads to tau fibril formation mainly in neurons, whereas CBD-tau injection results in predominantly oligodendrocyte-specific tau aggregation [108]. In another related study, conducted in non-transgenic wild-type mice, PSP-tau injection was shown to cause greater tau aggregation in the oligodendrocytes in the fimbria and corpus callosum with the potential for spatiotemporal tau spreading to interconnected regions of the brain [106]. 

These sorts of findings were further reproduced by another study, where insoluble fractions of tau isolated from CBD brains were stereotaxically injected into the right striatum of hTau mice (expressing all six human tau isoforms in a Tau KO background), after which significant phospho-tau inclusions (detected by CP13 antibodies recognizing Ser202) were observed in striatal gray matters and in the oligodendrocytes of the white matter tracts. More interestingly, prevalent spread of CBD-tau on the contralateral side of the brain was found in these mice, indicating potential transmission of these tau seeds in the brain parenchyma. The study also reported myelin degeneration occurring along with tau spreading through oligodendrocytes, as reduced MBP immunostaining was found in the corpus callosum after 12 months of CBD-tau injection. Again, such a CP13+ phospho-tau pathology was not prominent in the striatal astrocytes, raising the question of why oligodendrocytes are more vulnerable towards CBD-tau-induced neurodegeneration [109]. However, oligodendrocyte-mediated tau transmission strictly follows the routes enriched in oligodendrocyte cells and perhaps is independent of neuronal tau propagation. In an experimental study led by Ferrer and co-workers, it was shown that unilateral injection of the sarkosyl-insoluble fraction of tau isolated from human AD, PART, PSP, PiD, and FTLD-P301L brains into the corpus callosum of WT mouse brain produced oligodendrocyte-specific coiled bodies and threads of tau, which could even spread to the contralateral corpus callosum of the brain. Again, astrocytic tau accumulation as well as spreading was sparsely found in this study and that can be justified by the enriched presence of oligodendrocytes in the corpus callosum of the brain (75.4 ± 5.1% in human brain) [110,111]. Apart from stereotaxically injected mouse models of tauopathy, oligodendroglial tau pathology has been shown in other major studies performed in multiple transgenic animal models expressing human mutant forms of tau [112,113,114,115]. For example, mice expressing FTD with parkinsonism-linked mutated tau, G272V, under a prion promoter were shown to develop tau filaments having straight and twisted structures in oligodendrocytes. Aggregated tau filaments were also recognized by AT8 antibodies binding the Ser202/Thr205 motif of tau, suggesting pathological phosphorylation of tau. Tau pathology in oligodendrocytes and motor neurons of the spinal cord was evident in this mouse model, recapitulating human features of tauopathy [114]. In another study, employing rTg4510 mice expressing P301L-mutated human tau linked with FTD, researchers demonstrated the presence of filamentous mouse tau in the oligodendrocytes, although such inclusions were less in neurons [115]. Oligodendroglial tauopathy in these mice was accompanied by white matter degeneration marked with lower fractional anisotropy values in white matter tracts around the age of 8.5 months [116]. In the THY1-Tau22 mice expressing tau under the THY1 promoter, which is active in neurons but not in oligodendrocytes, researchers have shown age-dependent transfer of phospho-tau from the neurons to oligodendrocytes present in the white matter tracts of brain [117]. Furthermore, electron microscopy revealed swollen, degenerating axons and disrupted myelin sheaths [116,118], indicative of both oligodendrocyte and neuronal degeneration.

Along with the presence of oligodendroglial tau inclusions/filaments in tauopathy brain, it is also necessary to discuss the dysfunctions caused by tau aggregates on oligodendrocytes and their effect on neuronal integrity or degeneration in the brain. Studies conducted in a transgenic mouse model of tauopathy expressing the familial FTD-related P301L form of mutated tau in oligodendrocytes demonstrated early loss of axonal transport and significant myelin loss in the spinal cord, which was followed by generation of Thio S-positive aggregated tau and axonal degeneration, resulting in muscle atrophy and motor impairments. This form of oligodendrocyte-specific aggregated tau resembles the structures observed in human FTD brains, where disruption of myelin in the frontal white matter has been documented. The findings strongly support that disease specific abnormal expression of tau solely in oligodendrocytes can compromise neuronal functions in the CNS to cause neurodegeneration [103]. In another study, conducted in 3xTg-AD mice harboring human presenilin-1 M146V (PS1M146V), human amyloid precursor protein Swedish mutation (APPSwe), and the P301L mutation of human tau (tauP301L) [13], it was found that expression of oligodendrocyte markers like MBP and CNPase was markedly decreased in the CA1 area of the hippocampus and layer II/III of the entorhinal cortex of 6-month-old 3xTg mice compared to age-matched non-Tg mice. However, at that time point no significant loss of neuronal axonal markers or neurofilament protein was observed in these brain regions. This suggests that oligodendrocyte degeneration might precede axonal degeneration in tauopathy-containing mice [119]. This scenario has also been manifested in a recent study conducted in 6-month- and 24-month-old 3xTg-AD mice, where age-dependent loss of MBP immunostaining was found in the hippocampus of the 3xTg mice compared to the age-matched controls. In addition, the study reported early loss of self-renewal capacity of oligodendrocyte precursor cells (OPCs) in hippocampal subregions by 6 months of age and that sustained even up to 24 months of age concomitant with hypertrophy or unusual expansion of OPC volume, marking the abnormal functions of OPC in age-dependent neurodegeneration of tauopathy [120]. Collectively, these reports from experimental animal studies firmly establish dysfunction and chronic degeneration of oligodendrocytes occurring in the presence of tau pathology and imparting a passive and deleterious effect on neuronal physiology and functioning in the brain. 

### 4.2. Oligodendrocyte Dysfunction in Human Tauopathy

The findings from animal studies are further corroborated by the evidence obtained from human studies. Oligodendrocyte dysfunction and loss of white matter integrity are well evidenced in human postmortem brains of AD and other tauopathies, adding clinical relevance to the findings obtained from different transgenic mice lines. Using magnetic resonance imaging (MRI) it has been shown that white matter hyperintensities and abnormalities are related to cognitive decline in late-stage AD patients [121]. This correlates well with the presence of Aβ1-42 in the cerebrospinal fluid (CSF) and induction of tau level in the CSF, and therefore, white matter abnormality begins several years ahead of the clinical manifestation of AD in humans [122,123]. In human postmortem AD cortex, the number of Olig2+ cells were found to be decreased, and that is contrary to the APP/PS1 animal model of AD as the number of Olig2+ cells was increased in 6–8-month-old mouse brain, indicating repair mechanisms adopted by the OPCs to compensate for the myelin loss [124]. However, the compensatory mechanisms might be defeated at the late stage of disease, as evident in human brains. Moreover, the nuclear diameter of the OPCs in the parahippocampal white matter regions of human AD brains was found to decrease significantly, suggesting a possible degeneration of these cells in demented brains [125]. The recent single-nuclear-RNA sequencing (sn-RNA seq)-based analyses of confirmed human AD brains and the age-matched controls shed more light on the changes in the transcriptomic landscape of oligodendrocytes. The genomics analyses from human brain tissues identified five oligodendrocyte clusters distinguished by the nature of enriched transcripts. These clusters of oligodendrocytes were shown to express genes involved in different cellular pathways including glial cell development, apoptosis, synapse assembly and formation, cholesterol metabolism, and antigen processing/inflammatory pathways. Among all the clusters, the maximum of the differentially regulated genes was found in a single cluster, which demonstrated downregulation of genes involved in synapse formation, cell adhesion, vesicle transport, and ion transport through membranes, suggesting a collapse of the oligodendroglial functions in connecting with neuronal axons to form myelin sheath and other pivotal supportive functions. Along with that, some populations of oligodendrocytes were found to upregulate genes involved in cholesterol metabolism such as FMO5 and FDFT1 and downregulate genes involved in fatty acid synthesis (stearoyl-CoA desaturase), which are indispensable for the synthesis of myelin in oligodendrocytes. The upregulation of cholesterol synthesis genes by oligodendrocytes is attributed to another repair mechanism to resist the loss of myelin composition in the aged diseased brains [126]. Similar snRNA-seq studies conducted by previous researchers in human postmortem AD brains also documented upregulation of specific genes undergoing cholesterol metabolism and loss of expression of several genes contributing to synapse assembly and neuroprotective pathways [127,128], strengthening the occurrence of oligodendrocyte degeneration in AD. Along with AD, white matter degeneration is manifested in brains of FTD, CBD, PSP, PiD, and other forms of tauopathies, compromising the myelin formation and electrophysiological parameters of neurons. White matter degeneration has been confirmed in early pre-symptomatic as well as symptomatic phase of FTD [129,130,131] and the degeneration correlates well with the behavioral outcomes in patients [132]. Similarly, white matter abnormalities in the corpus callosum, superior cerebellar peduncle, cingulum, and uncinate fasciculus have been documented in PSP patients and shown to be strong predictors of overall disease severity in patients [133]. 

Neuropathological studies in preclinical and late-stage human CBD brains confirmed the presence of tau coiled bodies in oligodendrocytes present in the cortical white matter, striatum, and lentiform nucleus although such lesions were found to be less than the overwhelming astrocytic tau pathology [134]. This aligns with previous imaging-based comparative studies in postmortem CBD and PSP brains, where significant white matter pathology and volume loss were found in the frontal and parietal cortex and the basal ganglia of all the brain samples analyzed [135]. It can be stated that the findings of human tauopathy brains in terms of oligodendrocyte and overall white matter pathology are more comprehensive and demonstrate the varied region-wise degeneration of the diseased brain in contrast to the experimental animal models, which are restricted to certain regions of interest controlling either cognitive or movement functions of the animals. Therefore, the common pathological hallmark findings obtained from both animal and human studies are of great significance to dig deeper into the mechanisms of tau-induced oligodendrocyte degeneration and malfunction in tauopathy.

## 5. Conclusions

The present review succinctly describes the structural variants of tau and its aggregated forms found in human tauopathies. In addition, it provides special emphasis on elaborating the functional relation between tau and oligodendrocytes in normal and diseased brains based on animal and human studies. It is evident that disease-associated tau variants have different potentials for aggregation in selective brain cells, where neurons are found to be affected by both 3R and 4R tau, whereas oligodendrocytes are mainly susceptible to tau inclusions composed of 4R tau [136,137,138]. In addition to that, microglia and astrocytes have differing capacities depending on their activation states in influencing tau deposition and spreading in vulnerable brain regions. Furthermore, reactive gliosis contributes to enhancing the detrimental effect of tau propagation associated with dampening of synaptic plasticity and neuronal health, as shown in several studies [139,140,141,142,143]. On the other hand, oligodendrocytes are the pivotal components that form myelin sheaths, facilitate propagation of electrical signals, and provide growth support to neurons. Undoubtedly pathological mutations and aggregation of tau significantly disrupts these physiological functions of oligodendrocytes, and thereby, imparts deleterious effects on gross neuronal functioning, causing impairment of cognition, movement, and other brain-regulated functions in a context-dependent manner. Currently, there is a great lack of understanding about diverse tau-induced pathological mechanisms in oligodendrocytes. It is imperative to explore if oligodendroglial tau pathological mechanisms align or differ from that of the intraneuronal tau pathology. For example, mitochondrial dysfunction, oxidative stress, iron dyshomeostasis, lipid peroxidation, failure in protein degradation pathways, and inflammation induced by aggregated tau species within oligodendrocytes in tauopathy brain demands extensive investigations. Therefore, the present review raises the need for further in-depth studies on tau pathology in oligodendrocytes that will enrich our current knowledge and benefit future designing of novel tau-targeted therapeutic strategies to prevent human tauopathies.

## Figures and Tables

**Figure 1 cells-13-01112-f001:**
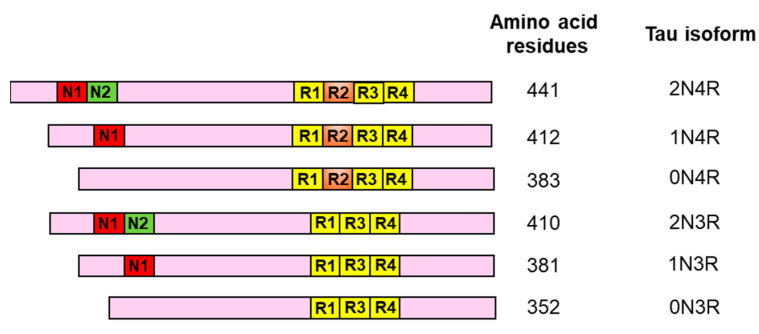
Human tau isoforms formed by alternative splicing of tau mRNA in brain: The figure demonstrates six different isoforms of human tau produced by alternative splicing of the tau mRNA. These variants are distinguished by the presence or absence of two N-terminal proline rich domains (N1 and N2) and four microtubule binding repeat domains (R1, R2, R3 and R4) towards the C-terminal of the protein. It represents the name of these isoforms namely 2N4R, 1N4R, 0N4R, 2N3R, 1N3R and 0N3R along with the differing length of each of these isoforms in terms of the total amino acid residues including 441, 412, 383, 410, 381 and 352 amino acids respectively.

**Figure 2 cells-13-01112-f002:**
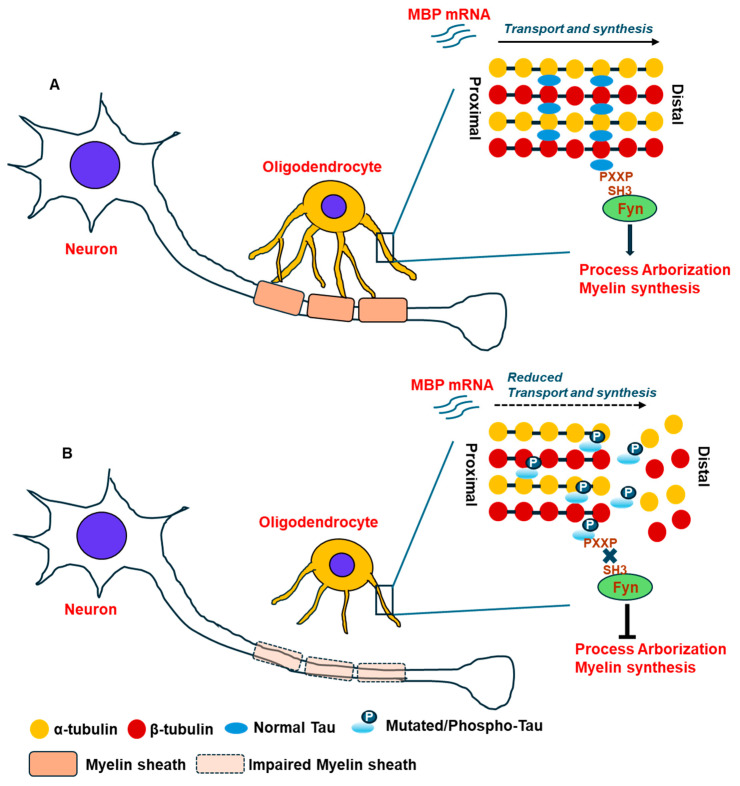
Function of tau in axonal transport and myelin sheath formation by oligodendrocytes in normal and tauopathy brains: (**A**) The diagram shows the integral function of tau in assembling the tubulins of the microtubule machinery and in interacting with the FYN kinase that promotes axonal transport of mRNA myelin basic protein (MBP) towards the distal end of the processes and collectively these phenomenon facilitates oligodendrocyte maturation, arborization, formation of myelin sheath over the neuronal axons in normal healthy brain. (**B**) Presence of mutated and/or abnormally modified tau by post-translational modifications (phosphorylation) causes diminished affinity to bind microtubule structural components and its interaction with the FYN kinase resulting in collapse of the microtubule and reduced transport of important cargos including mRNA of MBP towards the end of the processes. It leads to reduced arborization and myelin synthesis by oligodendrocytes and that causes breakdown of the myelin sheath over the neuronal axons as observed in diseased tauopathy brain.

**Table 1 cells-13-01112-t001:** **Post-translational modifications of tau:** The table lists major post-translational modifications of tau protein reported by numerous studies in animal and human brains. It includes the enzyme-mediated modifications of the target amino acid residues by phosphorylation, acetylation, nitration, ubiquitination, sumoylation, O-GlcNAcylation, and truncation. In addition, the table discusses the functional impact of these post-translational modifications in tau pathology and neurodegeneration.

Post-Translational Modifications	Target Amino Acid Residues	Enzymes Involved	Functional Impact	References
Phosphorylation	1. Thr-Pro and Ser-Pro motifs (85 known until now). 2. Tyr residues (5 identified until now).	Microtubule affinity-regulating kinases (MARKs), cAMP-dependent protein kinase (PKA), Ca2+- or calmodulin-dependent protein kinase II (CaMKII), glycogen synthase kinase-3β (GSK-3β), SRC family protein kinases (LCK, SYK and FYN), ABL family protein kinases (ARG and ABL1).	1. Phosphorylation in repeat domains and flanking regions (Ser262, Thr231, etc.) reduces its affinity to microtubules. 2. Phosphorylation at Ser202-Thr205 enhances its self-aggregation and NFT formation, whereas impairs axonal transport. 3. Phosphorylation decreases Tau–FYN interaction and impairs sorting of myelin proteins and neuron–glia interactions. 4. Phosphorylation reduces tau cleavage by other proteases (caspases).	[5,32,41,42,43,44,45,46,47]
Acetylation	Lys residues in the repeat domains and flanking regions of tau. More than 20 Lys residues in the full-length tau were found to be acetylated.	Acetylation by P300 acetyltransferase or by CREB-binding protein and deacetylation by sirtuin 1 (SIRT1) and histone deacetylase 6 (HDAC6).	1. Acetylation at Lys163, Lys280, Lys281, or Lys369 prevents its degradation. Acetylation at Lys274 and Lys281 reduces its binding to microtubules. Lys274 acetylation is found in NFTs of AD brain and correlates with disease pathology. 2. Acetylation at Lys259, Lys290, Lys321, or Lys353 promotes its degradation and reduces phosphorylation. 3. Acetylation at certain Lys residues blocks ubiquitination and promotes formation of more tau oligomers and aggregates resulting in synaptic degeneration and cognitive deficits.	[48,49,50,51,52,53,54]
Nitration	The Tyr amino acids present in the structure of tau have been shown to be nitrated (addition of -NO_2_). Some of these nitrations (Tyr197) happen in healthy brains, whereas some nitrations (Tyr18, Tyr29, Tyr394) are found in AD brains.	Nitration of tau, like other proteins associated with neurodegeneration, is mainly caused by reactive nitrogen species such as peroxynitrite formed because of nitrosative stress.	Nitrated tau was shown to have reduced binding with microtubule in vitro. Nitrated tau has been found in NFT-bearing neurons and in glial tau inclusions.	[55,56,57,58,59]
Ubiquitination	A total of 17 Lys residues in the 2N4R form of tau are known to be ubiquitinated and majority of these Lys residues are buried in the microtubule-binding domain.	Multiple E3 ubiquitin ligases such as C-terminus of the Hsc70-interacting protein (CHIP), the TNF receptor-associated factor 6 (TRAF6), and axotrophin/MARCH7 have been reported to ubiquitinate tau through Lys 48- and Lys 63-dependent mechanisms.	1. Mono- and poly-ubiquitination of tau at distinct Lys residues drive its proteasomal degradation in cells. 2. Phosphorylation of tau may precede ubiquitination and promotes or inhibits ubiquitin-mediated degradation of insoluble tau depending on the site of Lys residue in the tau protein. Therefore, ubiquitinated tau is also found in the pathological tau inclusions and NFTs in tauopathy brain. 3. Soluble tau is shown to be ubiquitinated through Lys 63 modification that leads to autophagy-mediated clearance of tau. 4. In addition, chronic impairments of the ubiquitin–proteasome system further upregulates tau inclusion formation and pathology.	[60,61,62,63,64]
Sumoylation	Small ubiquitin-like modifiers (SUMOs) are bound to Lys residues of tau inhibiting ubiquitin binding and proteolysis of tau.	Sumoylation is carried out by E1 activating enzyme, E2 conjugating enzyme, and E3 ligase of the ubiquitination machinery.	1. Sumoylation at Lys340, present in the microtubule-binding domain, promotes hyperphosphorylation of tau and inhibits ubiquitin-mediated degradation, leading to enhanced aggregation of tau. 2. Sumoylation also hinders tau clearance by the autophagy-lysosome pathway, as shown in the rTg4510 mice model of tauopathy.	[57,65,66,67]
O-GlcNAcylation (O-GlcNAc)	Ser-Thr hydroxyl groups that are prone to be phosphorylated are also O-glucosylated by N-acetylglucosamine.	O-GlcNAc transferase (OGT) transfers the addition of N-acetylglucosamine to serine or threonine hydroxyl groups of tau protein. O-GlcNAcase (OGA) removes the N-acetylglucosamine group from tau. OGT is transcriptionally upregulated by CREB binding to its promoter (CRE site). In contrast, SIRT1-mediated deacetylation of CREB reduces OGT transcription and inhibits O-GlcNAc of tau.	1. O-GlcNAc inhibits phosphorylation and pathological aggregation of tau. Higher O-GlcNAc of tau has been shown to protect brain from tau-induced impairments. 2. O-GlcNAc of tau is decreased in AD and perhaps caused by reduced glucose uptake and metabolism in AD brains.	[68,69,70,71]
Truncation	Around 60 cleavage sites have been identified in full-length tau and several of these truncated forms are found in AD brains.	Disintegrin and metallopeptidase domain 10, asparagine endopeptidase (AEP), calpains, caspases, cathepsins, chymotrypsin. In addition, autoproteolysis of tau occurs at K281-L282 and K340-S341 when acetylation happens on Lys residues.	1. The cleavage of tau may expose the microtubule-binding domains and facilitate its self-aggregation. Truncated tau fragments including E391- or D421-truncated Tau, Tau304–380 fragment, K18/K280, and K12 aggregate without the presence of negatively charged inducers. 2. Cleaved Tau N368 fragments are more abundant in animal models of tauopathy while colocalizing with NFT. 3. Deletion of the first 150–230 amino acids of tau results in more phosphorylation and aggregation. Similarly, deletion of the last 50 amino acids of tau causes enhanced phosphorylation and self-aggregation.	[72,73,74,75,76,77,78,79,80]

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
