# Peer review of "Oligodendrocyte Dysfunction in Tauopathy: A Less Explored Area in Tau-Mediated Neurodegeneration"

_cells, 2024, doi:10.3390/cells13131112_

Round 1
Reviewer 1 Report
Comments and Suggestions for Authors
I have read with interest the manuscript titled “Oligodendrocyte dysfunction in Tauopathy: A less explored area in tau-mediated neurodegeneration.” by Majumder and Dutta. The authors cover a field that has been slightly overlooked in research of neurodegenerative diseases, e.g. the contribution of oligodendrocyte dysfunction in tauopathies. I enjoyed reading the manuscript and I believe that it is filling an important gap summarising the reported knowledge of pathology in other cell types apart from neurons. I only spotted a couple of spelling mistakes (see bellow), but otherwise I endorse the publication of the review article.
Line 229: spelling mistake nfigeural axons
Line 274: a dash is needed between “brain derived”
Comments on the Quality of English Language
Only a couple of spelling mistakes:
Line 229: spelling mistake nfigeural axons
Line 274: a dash is needed between “brain derived”
Author Response
Comments: I have read with interest the manuscript titled “Oligodendrocyte dysfunction in Tauopathy: A less explored area in tau-mediated neurodegeneration.” by Majumder and Dutta. The authors cover a field that has been slightly overlooked in research of neurodegenerative diseases, e.g. the contribution of oligodendrocyte dysfunction in tauopathies. I enjoyed reading the manuscript and I believe that it is filling an important gap summarising the reported knowledge of pathology in other cell types apart from neurons. I only spotted a couple of spelling mistakes (see below), but otherwise I endorse the publication of the review article.
Line 229: spelling mistake nfigeural axons
Line 274: a dash is needed between “brain derived”
Response: We humbly thank the reviewer for enjoying reading this review article and acknowledging the importance of this overlooked topic. We are also thankful for identifying the spelling mistakes. Accordingly, we have corrected the spellings of the words in these two mentioned lines and highlighted these in the submitted version.
Reviewer 2 Report
Comments and Suggestions for Authors
This review is dedicated to structural variants of tau and its aggregated forms found in human tauopathies. n addition, it provides special emphasis on elaborating functional relation between tau and oligodendrocytes in normal and diseased brains based on the animal and human studies. The review is interesting, informative, based on a large body of studied and elaborated literature. The review is well planned, the figures presented reflect what is written in the text. I recommend this material for publication
Author Response
Comments: This review is dedicated to structural variants of tau and its aggregated forms found in human tauopathies. n addition, it provides special emphasis on elaborating functional relation between tau and oligodendrocytes in normal and diseased brains based on the animal and human studies. The review is interesting, informative, based on a large body of studied and elaborated literature. The review is well planned, the figures presented reflect what is written in the text. I recommend this material for publication.
Response: We are grateful to the reviewer for the positive comments and for endorsing this review article for publication.
Reviewer 3 Report
Comments and Suggestions for Authors
Review: refers to manuscript entitled “ Oligodendrocyte dysfunction in Tauopathy: A less explored 2 area in tau-mediated neurodegeneration” by Moumita Majumder and Debashis Dutta
This is an interesting and well-written review paper on the role of tau proteins in oligodendrocytes in the normal state and in pathological conditions. Over the years, considerable attention has been paid to the neuronal mechanisms of tauopathy, whereas studies on oligodendrocytes in tauopathy-associated neurodegeneration have been less explored. Therefore, the collection of information on the study of tau function in oligodendrocytes and the consequences of disrupting these functions is much needed and valuable. The authors' paper highlights the striking contrast between the vast amount of research on tau in neurons and the much more sparse information on tau in glial cells. It is also valuable to compare the available data on tau and oligodendroglia activity obtained from studies in animal models with the clinical data obtained in humans.
It appears that in oligodendrocytes, as in neurons, a key role for tau protein is its involvement in mechanisms of intracellular transport and communication between nerve cells. Also, in a number of pathological disorders, key processes that are destroyed in both neurons and oligodendrocytes are related to transport mechanisms.
Throughout the manuscript, there are only a few things that cause some concern.
1. In Keywords: What is the difference between tau and MAPT?
2. Lines 110-113: These sentences unnecessarily repeat the information contained above and which is also described in Table 1.
3. Lines 161-173: Why is such elaborate information on Pick's disease to be used? It does not seem to be needed in this form.
4. One more general comment. The first chapters of the manuscript contain information that, however useful, has all been presented many times in other reviews or textbooks and does not in itself say anything about the relationship between tau and oligodendroglia.
Author Response
This is an interesting and well-written review paper on the role of tau proteins in oligodendrocytes in the normal state and in pathological conditions. Over the years, considerable attention has been paid to the neuronal mechanisms of tauopathy, whereas studies on oligodendrocytes in tauopathy-associated neurodegeneration have been less explored. Therefore, the collection of information on the study of tau function in oligodendrocytes and the consequences of disrupting these functions is much needed and valuable. The authors' paper highlights the striking contrast between the vast amount of research on tau in neurons and the much more sparse information on tau in glial cells. It is also valuable to compare the available data on tau and oligodendroglia activity obtained from studies in animal models with the clinical data obtained in humans.
It appears that in oligodendrocytes, as in neurons, a key role for tau protein is its involvement in mechanisms of intracellular transport and communication between nerve cells. Also, in a number of pathological disorders, key processes that are destroyed in both neurons and oligodendrocytes are related to transport mechanisms.
Throughout the manuscript, there are only a few things that cause some concern.
Comment 1. In Keywords: What is the difference between tau and MAPT?
Response: We are thankful to the reviewer for pointing out the repetition of tau and therefore we have deleted it from the keywords. Instead, we have added ‘neurodegeneration’ as another keyword.
Comment 2. Lines 110-113: These sentences unnecessarily repeat the information contained above and which is also described in Table 1.
Response: As per reviewer’s suggestion we have deleted the information contained in lines 110-113 and just stated that “Varied forms of post-translational modifications of tau along with the causing proteins/enzymes are described in Table 1” (highlighted in the text).
Comment 3. Lines 161-173: Why is such elaborate information on Pick's disease to be used? It does not seem to be needed in this form.
Response: We genuinely thank the reviewer for raising this important question. We would like to provide two references, Ikeda et al., 1995, Acta Neuropathologica and Richter-Landsberg and Bauer, 2004, Int J Dev Neurosci, where it has been stated that oligodendroglial tau inclusions are prominent tau pathologies observed in Frontotemporal dementia, including Pick’s Disease. Therefore, it was imperative to discuss the structural differences of tau aggregates formed by the 3R forms of tau in Pick’s disease to distinguish it from other forms of tauopathies and to relate if this structurally diverse form of tau aggregates can also affect oligodendrocytes to the similar extent as found in 3R+4R tauopathy and only 4R tauopathy.
Comment 4. One more general comment. The first chapters of the manuscript contain information that, however useful, has all been presented many times in other reviews or textbooks and does not in itself say anything about the relationship between tau and oligodendroglia.
Response: We humbly thank the reviewer for raising such an important point. The aim of this review is to describe the role of tau oligodendrocyte maturation/function etc. and its pathological alteration under diseased conditions like in tauopathy brains. In this course it was important to elaborate some basic and useful information about the different splicing variants and conformations of tau protein and its disease specific aggregated forms as different conformers of tau demonstrate differing proclivity to damage specific cell types in brain. Although majority of these information, as discussed in the initial part of the review, have been reiterated in other reviews published by established investigators in this field, we thought to briefly elaborate it as an introductory element to provide a general knowledge about tau and its disease specific pathological forms and that may help the readers understand the basic biology of this protein before articulating to the main content of the article, which is the function of tau in oligodendrocyte maturation and its impact under diseased conditions. We hope that the organization of the overall content of this review is scientifically relevant and adheres to the criteria of this journal.